
# The catastrophe of the Niedów dam - the causes of the dam's breach, its development and consequences

Stanisław Kostecki[1,*] and Robert Banasiak[2,*]

[1]Faculty of Civil Engineering, Wrocław University of Science and Technology
[2]Institute of Metheorology and Hydrology and Water Management – National Research Institute

**Correspondence:** Robert Banasiak (Robert.Banasiak@imgw.pl)

**Abstract.**

Due to extreme rainfall in 2010 in the Lusatian Neisse River catchment area, a flood event with a return period of over 100 years occurred, leading to the failure of the Niedów dam. The earth-type dam was washed away, resulting in the rapid release of nearly 8.5 million $m^3$ of water and the flooding of the downstream area with substantial material losses. The paper

analyses the conditions and causes of the dam's failure, with special attention given to the mechanism and dynamics of the compound breaching process, in which the dam's upstreeam slope reinforcement played a remarkable role. The paper also describes a numerical approach for simulating a combined flood event along the Lusatian Neisse River with the use of a two-dimensional hydrodynamic model (MIKE21). The flood event occurred downstream from the dam. Considering the specific local conditions and available data set, an iterative solution of the unsteady state problem is proposed. This approach enables

realistic flood propagation estimates to be delivered, the dam breach outflow to be reconstructed, and several important answers concerning the consequences of the dam's failure to be provided.

## 1 Introduction

The number of dams for storing and supplying water is increasing worldwide due to the growing demand from towns, agriculture, industry, or power generation. Dams also play an important role in reducing the risk of flooding. Apart from the substantial

benefits to society provided by dams, there is also an inherent and growing risk of dam failure. This results in flooding that can cauese serious material damage and loss of life. The failure of the dam could have occurred due to faults during the design and construction stages, the aging of the structure, and climate change that resulted in the altering of meteorological and hydrological patterns (Grant, 2001; Hansen et al., 2020; Ho et al., 2017). The International Commission on Large Dams (ICOLD, 1995, 2011) has reported 176 failures among the 17,406 registered dams in the world. According to the Commission, the failure rate

for embankment dams is higher than for concrete dams. It also revealed, in the case of embankment dams, that overtopping failure is the most common cause of failure when compared with other types of failures like piping and slope failure. Analysis of the dams' failure plays a key role in understanding the mechanisms of such disasters (Wu , 2011). This in turn enables more accurate methods of forecasting failures, as well as ways to prevent them, to be developed. These actions are a great help for administration bodies when preparing flood hazard maps and contingency plans, which allow for a quick and effective response



to disasters (Alcrudo and Mulet, 2007). However, obtaining detailed data on the course of such an event is difficult, because on the one hand, the activities of the services in a hazardous situation come down to protecting people and valuable areas against a flood wave, and on the other hand, disasters occur unexpectedly and under the conditions of limited monitoring. Therefore, the analysis of the dynamics of a disaster and the development of a breach, as well as the preparation of the hydrograph of the outflow, are performed after the flood on the basis of often uncertain or scarce data. However, due to the importance of this

issue, efforts are made to recreate these events and expand the existing database of the descriptions of the causes and effects of recent disasters (Alcrudo and Mulet, 2007; Yochum et al., 2008; Azeez et al., 2020), as well as those that occured a long time ago, including those from the first half of the last century (Begnudelli and Sanders, 2007; Pilotti et al., 2011).

The key data for assessing the consequences of dam failure in terms of the downstream inundation time, depth of flooding, and extent of possible damage, is the outflow hydrograph. The hydrograph's shape, volume, and peak outflow depend on the

evolution of the dam break, the height of the dam, and the reservoir storage volume. Using regression formulas e.g. (Bureau of Reclamation, 1988; Froehlich, 2008; Xu and Zhang, 2009), the evolution of a dam breach can be assessed relatively easily and correctly, but only for simple cases, i.e. dams made of homogeneous soil. However, earth dam structures are typically more complex, and consist of a number of different material layers. They are equipped with sealing cores, drainage facilities, and wave protectors, and can even have a paved road on the top. Therefore, forecasting a breach in a dam is much more complicated,

and in most practical applications various simplifications and approximations are used (Kostecki and Rędowicz, 2014).

Another approach of calculating dam failure parameters involves numerical methods that use equations of fluid motion (such as Navier-Stokes or Saint-Venant equations) coupled with erosion extensions for the dam's body (eg. dam break module in MIKE11, DHI (2011)). The main advantage of these methods is the more precise consideration of the parameters describing the mechanical properties of the soil, such as cohesion, friction, etc. This means that if appropriate parameters are available,

the numerical method is more accurate than the empirical method (Saberi , 2016). However, in the case of a more complex dam stucture, the uncertainty of predicting its washing away also increases due to the difficulty of correctly determining the parameters of the model (Borowicz and Urbański, 2011). The evaluation of the consequences of dam failure relies on the testing of a number of catastrophic scenarios in order to further analyze and assess the consequences of the potential flood. The basis of such analysis includes hydrologic simulations, numerical modelling of breaching processes, flood plain

flows, and the preparation of inundation maps using GIS systems (Altinakar, 2008; Cleary et al., 2012, 2015; Cannata and Marzocchi, 2012; Álvarez, 2017; Zhong, 2011). 1D models can predict the flood propagation in channels and narrow valleys with reasonable accuracy and good efficiency (Pilotti et al., 2011; Teng et al., 2017; Tayefi et al., 2017). However, a 2D or hybrid1D/2D approach should be used in wide floodplains and complex terrain regions with elevated roads, secondary dikes, levees, buildings, and other obstacles (Vanderkimpen and Peeters, 2008). The 2D models are more commonly applied due

to significant computational power advances and the availability of air-born topographical data in recent years (Saberi et al., 2013; Yakti et al., 2018; Banasiak, 2021). In addition, hybrid models are used to derive 1D based breach outflow hydrographs, whereas the 2D models are used for flood plain modelling and the generation of inundation maps downstream of a dam (Shah et al., 2019).



The current work presents a case study of a catastrophic failure of the Niedów dam in Poland on the 7th of August, 2010.
The goals of the study are to give an explanation of the failure mechanism in the case of this concrete face earth dam and to
determine the impact of the failure on the flooding of the area below. This will enable it to be used as a case study for the
testing and validation of the breaching of similarly constructed dams. The geographic, meteo- and hydrological conditions
leading to this event are also presented. Finally, a 2D hydrodynamic model was applied in order to more reliably determine the
hydrograph of the outflow from the reservoir and the propagation of the flood wave for the purpose of determining the impact
of the failure on the flooding of the area below.

## 2  Study Site

### 2.1  Description of the study area

The Niedów dam on the Witka river (at km 2.2) is located in south-west Poland, near the Polish-Czech and Polish-German
borders. It was constructed in 1962 to supply water to the Turów coal power station for cooling purposes, and also to supply
drinking water to nearby settlements, including the town of Bogatynia. In essence, the function of the reservoir was not to
mitigate the flood hazard. The storage capacity of the reservoir before failure - at a normal water level of 210 m a.s.l. - was 5.6
million $m^3$, and the water surface area was 183 hectares. The reservoir's catchment area is 321 $km^2$ in the (sub)mountainous
region of the Izerskie Mountains, and has significant stream slopes. Most of the catchment area is located on the territory of
Czechia, as shown in Figure 1. The geological structure of the river bed is made up of granite and gneiss formations under a
layer of sands, gravels, and clay (locally). Such formations are favorable for high run-off.

### 2.2  Description of the Niedów Dam

The Niedów dam consisted of three major sections: the central section (with a concrete water release structure equipped with
movable gates, a bottom outlet and a hydropower plant), and two earth embankments. The total length of the central part of the
dam was equal to 47.05 m. The length of the earth embankments on the left and right side was 126 m and 94 m, respectively
(see Fig. 2).

Three tainter steel gates, with a width of 6.7 m and a height of 6.6 m each, controlled the water outflow from the reservoir
(see Fig. 3). The maximum yield of the weir, when the gates are elevated by 5.0 m and the water level in the reservoir reaches
210 m a.s.l., is 500,0 $m^3s^{-1}$. This corresponds to the design flow with an exceedance probability of 1%. This yield can reach
a value of 655 $m^3s^{-1}$ for the designed maximum water level of 210.4 m a.s.l. In addition, the pillars of the central section
contained bottom outlets with a size of 2 m x 1 m, which were equipped with vertically moving flat closures. The yield capacity
of each outlet was equal to 10 $m^3s^{-1}$ (at a water level upstream of 210 m a.s.l. and a water level downstream of 202,20 m
a.s.l.). In normal conditions, these openings were utilized to empty the reservoir. They were also used during the catastrophic
event.

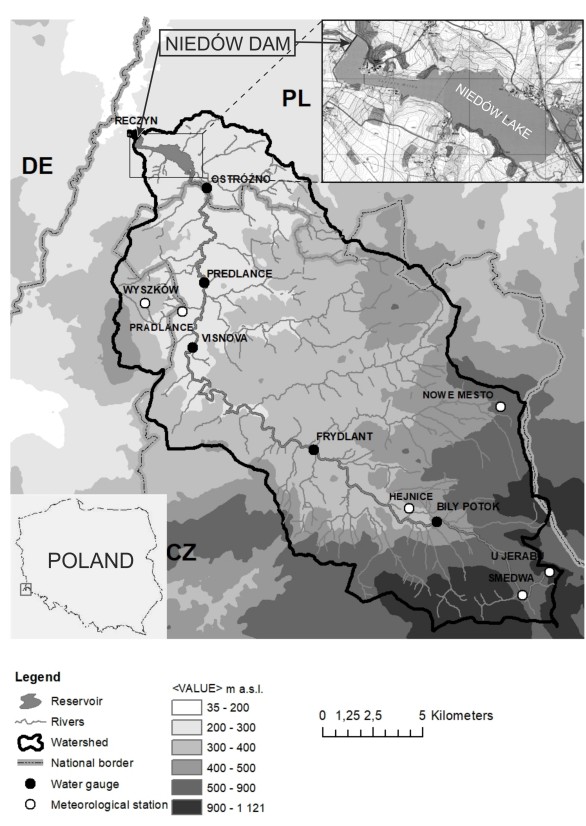

**Figure 1.** The catchment area of the Witka River

The structure and geometry of the earth dam are presented in Figure 4. The maximum height of the embankment with respect
to the base ground level was 11.6 m. The body of the dam was made of well compacted sand without a clay core. The slope
upstream was of a ratio of 1:3, while the slope downstream had a ratio of 1:2.5. Because the sand had a high permeability
coefficient of $2.8 \times 10\text{-}3$ ms$^{-1}$, the upstream slope was shielded with a double layer of concrete slabs with dimensions of
$1.5 \times 1.5 \times 0.1$ m, which were sealed with a bituminous material. The shield from the upstream water was supported by a vertical
reinforced concrete cut-off wall, which was reaching down to the basement rock. The downstream slope was covered with
humus and grass. In the lower part of the slope, there was a drainage of mixed gravel and stone. The dam's crest was 5 m wide
and served as a road made of concrete slabs with asphalt. The power plant and water outlet sections were connected to the dam
with abutments. The total volume of the earth dam was ca. 61,000. m$^3$

The dam was technically supervised regularly, and was stable and in good condition. A number of maintenance and restoration works were executed in the years from 1998 to 2009, including the repair of the steel and concrete structures, the repair
of the upstream slope, and the replacing of the road pavement on the top of the dam in 2009. The dam was operated according




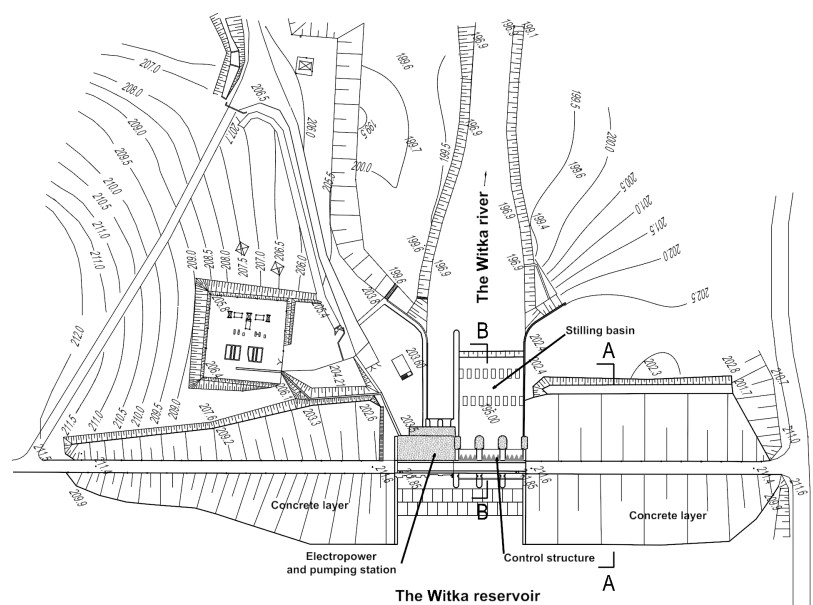

**Figure 2.** Plan view of the Niedów Dam

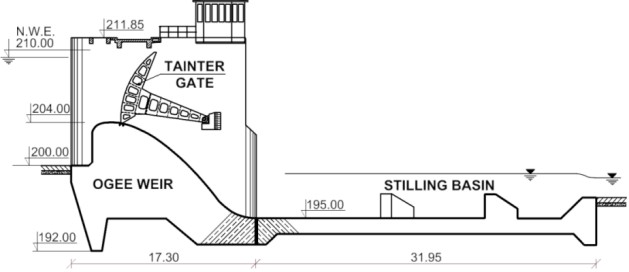

**Figure 3.** Ogee weir cross-section

to its documentation, which consisted of five major items: i) guidelines for the operation of the water intake, ii) guidelines for flood management for the reservoir area, iii) technical instruction of the dam's operation during a flood, iv) a manual for gate control, v) a manual for the operation of the power plant.

## 2.3 Meteorological and hydrological conditions

In the period between 6 and 8 of August, 2010, the upper catchment area of the Lusatian Neisse (in Polish - Nysa Łużycka) was subjected to exceptionally high amounts of rainfall. In the Witka catchment area, a tributary of the Lusatian Neisse cumulated rainfall reached values in the range of 150-250 mm in 48-hours, and the daily sum on the 7th of August reached values of 128.5 and 179 mm in the metheorological stations of Mnisek and Heinice, respectively (see Fig. 5). The most intensive




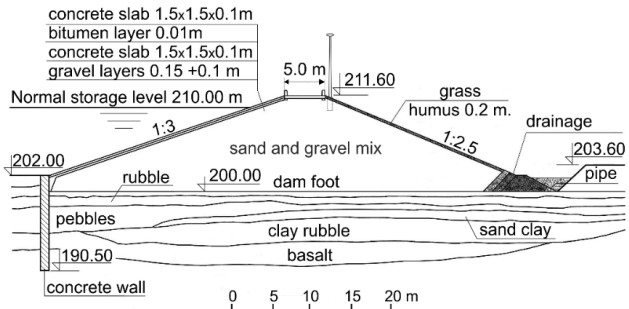

**Figure 4.** Cross-section of the earth dam

rainfall occurred in the morning between 8 and 9 a.m., with 15-35 mm of rain falling in an hour - locally reaching even 58
mm at the Heinice station. This rainfall statistic corresponds to one-fourth of the yearly rainfall in this mountainous region.
Moreover, the hydrological situation deteriorated due to the precedent wet period in the second half of July (with precipitation
above the norm), which led to the saturation of the ground and the acceleration of the subsequent run-off. The consequence of
such a meteorological situation was the occurance of catastrophic floods on several rivers, including flash floods on the Witka
River, the Miedzianka River, and the Lusatian Neisse River (IMGW et al., 2010). Figrure 6 shows a map that indicates the
hydrological network in question, and the meteorological and hydrological observation stations. In most hydrological gauge
stations, the observed water levels significantly exceeded the historical maxima. Remarkably, a number of gauge meters were
destroyed during the passage of the floodwater, making it more difficult to subsequently assess the quantitative data of the
flood. The return period of the flood is estimated to be within 100-200 years.

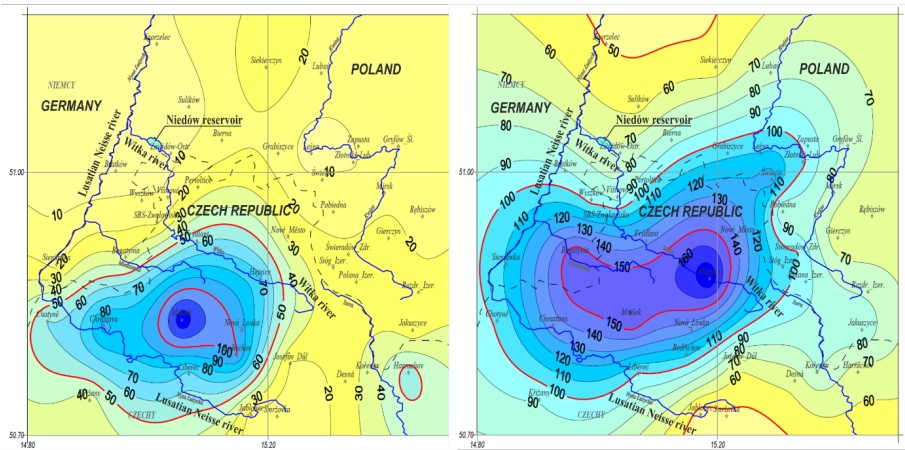

**Figure 5.** Rainfall values for 6.08.2010 (left) and 7.08.2010 (right) (in mm)

The flow of the Witka river - its name on the Czech territory is the Smeda river - is monitored at four gauge stations. On the
Polish section from km 0.0 (river mouth) to km 8.0, there are two stations: Ostróżno (km 7.98) - upstream from the reservoir,


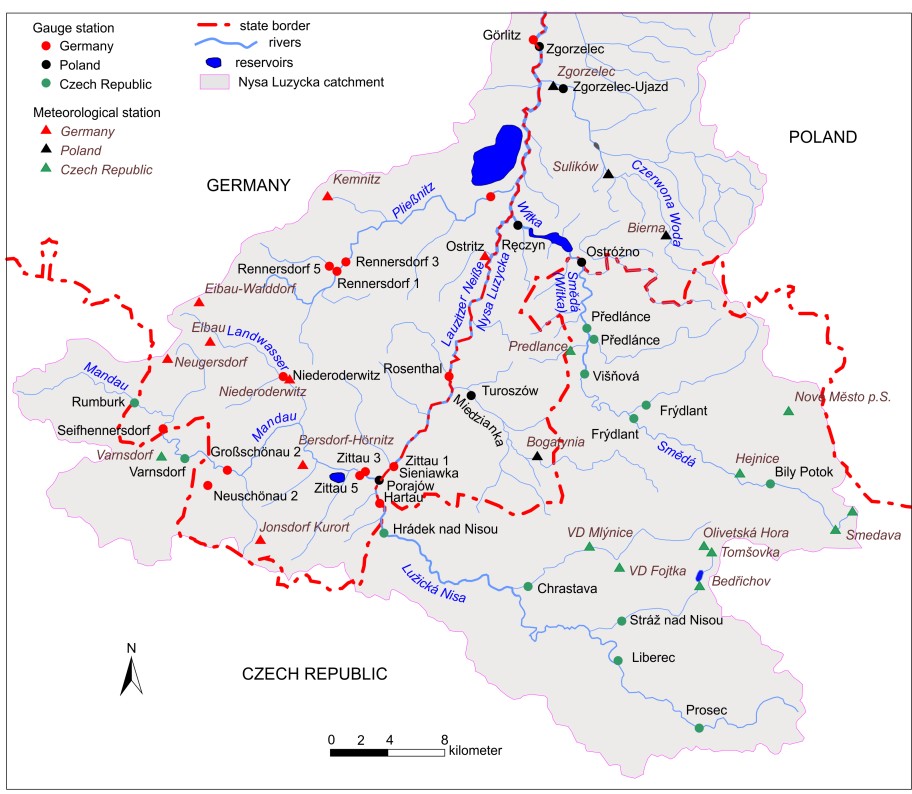

**Figure 6.** The upper Lusatian Neisse catchment area up to the Görlitz (GE)/Zgorzelec (PL) gauge station (source: IMGW et al. (2010))

and Ręczyn (km 1.8) - downstream from the reservoir (Fig. 2). On the 7th of August at the Ostróżno gauge station, the highest water level of the flash flood occurred at 16:40. The Ręczyn gauge station was recording the water level until the time of 15:20, and thus until it was destroyed due to the high release of water from the reservoir before the failure of the dam. During the 45-year period of continuously monitoring flow at Ostróżno gauge station, the flood discharges were less than $70\,\mathrm{m^3s^{-1}}$, which
is still within the limit of bankful flow. There was only one case of higher flow, which occured in August, 2001 and which was equal to $171\,\mathrm{m^3s^{-1}}$. That event also featured a rapid ascent and descent of the wave, which is typical for a flash flood. On the 7th of August, the estimated flood rate was $615\,\mathrm{m^3s^{-1}}$, but this estimation is still burdened with significant uncertainty. On the 7th of August, the estimated flood rate was 615 m3s/1. This estimation was difficult, as the water level substantially exceeded the measured range and due to locally wide floodplain. Between the Ostróżno cross-section and the reservoir, there
is an increase of the catchment area from 268 to 331 $\mathrm{km^2}$, including the Koci Potok stream, which also severely flooded and delivered a significant direct inflow to the reservoir. This stream was not monitored, but based on the field survey after the flood, the peak flow rate was estimated at ca. $70\,\mathrm{m^3s^{-1}}$.



### 2.4 Dam failure - wash out mechanism and breach characteristics

The water level in the reservoir was controlled according to the operational manual. The procedure involved the gradual
elevation of the gate by 0.2 m in order to maintain the desired water level. When the control of one gate was insufficient,
the additional gate was also raised by 0.2 m. In the course of this unpreceded water level rise, the opening of the gate was
accelerated. During the catastrophic flood, one of the two turbines was undergoing renovation. After the water level exceeded
the edge of the repaired gate, the water flowed into the hydroelectric power plant. As a result, the control room was also flooded,
the crew was evacuated to the top of the dam, and the power supply was turned off. The crew still tried to open more gates
manually from the dam's crest, but were unsuccessful. The overflow started at 17:00 over the left side of the dam near the
bank because the crest was slightly inclined towards it. The water passing over the crest caused the erosion, which first occured
around the lamp post's foundations, and then on the slope covered with grass, which is documented in Sup. 1. This process took
about half an hour and resulted in the gradual disintegration of the road on the dam's crest. The concrete slabs then lost their
support and fell due to the washing away of the sand that constituted the dam's body. Remarkably, the concrete slabs, when
losing support, broke in a series like chocolate, and were swept away by the intensified flow. Afterwards, another important
moment occurred. As the support of the earth embankment vanished, the left training wall flanking the central concrete dam
collapsed due to upstream water pressure. This resulted in a further rapid outbreak. This phase was relatively short but intense,
and resulted in a torrential flood wave downstream, which is documented in Figure 7. After the next 80 minutes, the left side
of the earth dam was almost completely swept away (Fig. 8).

The overtopping of the right dam began approximately 15 minutes after the left one. The breaching in this case developed
in a similar, but less dynamic way. The washout started at the central part of the right dam, and evolved towards the right bank
of the valley. As a result of the fall of the left abutment, and a fast lowering of the water level in the reservoir, the right bank
washing out decelerated. In addition, the concrete slabs resisted failure and worked as a weir for about 20 minutes. It is difficult
to explain the origin of this. Possibly, the slabs jammed, or concrete debris temporally hindered the erosion process. Figure
7 shows the hindered breaching of the right side of the dam. Finally, the concrete slabs were washed away. Nevertheless,
the outflow here was not that intense, since the upstream water level had already substantially decreased. The washing out
of the right side of the dam lasted for about 130 minutes, causing the devastation of 62 per cent of its length. The washing
out reached the level of 200.00 m a.s.l (the bottom of the breach). The final width of the breach of the right dam was 58 m.
Figure 9 illustrates the complete dam breach. Part of the dam adjacent to the control structure remained. Table 1 collects the
crucial moments of the development of the breach, established on the basis of observations, records, and interviews. The breach
parameters, i.e. the eroded earth volume $V_{er}$, the mean breach width $B_{avg}$, and breaching time $T_f$ are further collected in Table
2 for the left and right embankment.

### 2.5 Flooding downstream the Niedów dam

At 4 p.m., the high water discharges from the Niedów reservoir caused an overflow of water from the banks of the Witka
River, and also the flooding of the adjacent areas. At 4:20 p.m., the Ręczyn water gauge station on the Witka River (km 2.2)



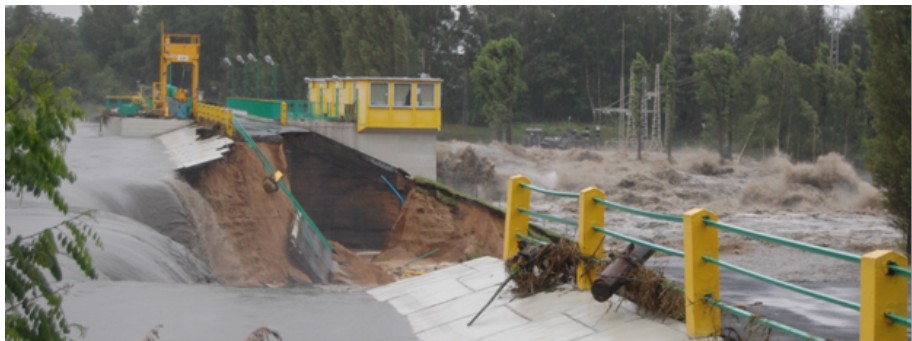

**Figure 7.** View from the right bank. Behind the concrete structure, an immense outflow is visible after the abutment had collapsed. Water level in the reservoir ca. 210.60 m a.s.l.

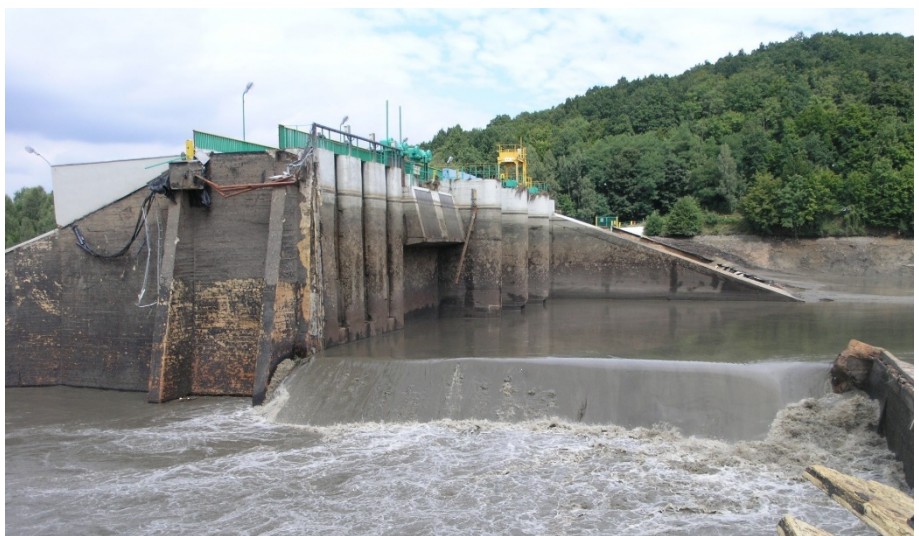

**Figure 8.** Broken left retaining wall of the control structure

was destroyed, and the flood waters headed to the mouth of the river into the Nysa Łużycka River and towards the village of Radomierzyce through the Mill channel. The dramatic situation occurred at 17:20 when the left dam broke. The rapid outflow of water destroyed the weir on the Witka River (which directed the water to the Mill channel) and the railway line on the embankment. The wave then reached the nearest village of Radomierzyce (cf. Fig. 11). The areas of the village were flooded
to a depth of about 2 meters. However, there was no destruction (disintegration) of the buildings. The flood wave of the Witka River, combined with the flood wave of the Nysa Łużycka River, caused the flooding of the areas on the German side of the border. First, it flooded the Hagenwerder estate, and then the water overflowed over the embankments into the Berzdorff reservoir, destroying the railway line running along the border. The subsequent serious flood damage took place on August 8, between 00:00 to about 12:00 on the further section of the Nysa Łużycka River when the wave reached the city of Görlitz on




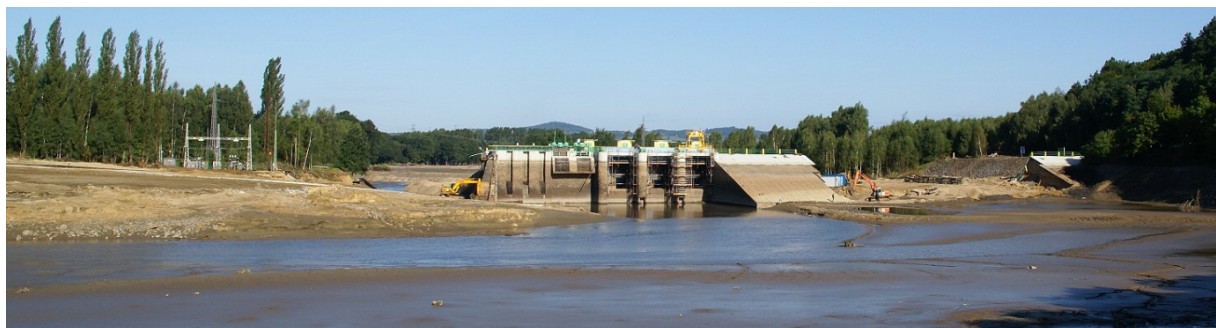

**Figure 9.** The final breach of the dam - an upstream view

**Table 1.** The development of the failure of the Niedów dam on August 7, 2010

| Time | Development |
|------|-------------|
| 15.00 | Outflow from the reservoir - 86 $\mathrm{m^3s^{-1}}$, WL - 210,02 m a.s.l., gates I, II and III open 60 cm. |
| 15.36 | Water inflow into the power plant and in the control room, crew evacuation, and manual opening of the gates. |
| 15.50 | Outflow 140 $\mathrm{m^3s^{-1}}$, WL 210,21 m a.s.l., gates I, II and III open 150 cm. |
| 16.10 | A rapid rise of the water level in the reservoir. |
| 16.40 | The maximum water level at the Ostróżno gauge station. |
| 17.00 | Beginning of the flow over the left side of dam, the crew evacuated, gates I and III open 250 cm, II open 200 cm, outflow 352 $\mathrm{m^3s^{-1}}$. |
| 17.15 | Beginning of the flow over the right side of the dam. |
| 17.42 | Water level reaches a maximum of 212.05 m a.s.l. Washing out of the downstream slope of the dam, destruction of the road on the top of the dam – dam breach width of ca. 40 m on the left side and 30 m on the right side. |
| 18.10 | Water level at 211.60 m a.s.l. The breaching probably reaches the dam's floor. The collapse of the headwall of the left side of the dam resulted in the immense outflow through the breach. |
| 18.47 | The breach of the left side of the dam finishes, and has a length of 106 m. |
| 18.56 | Water level - 209.00 m a.s.l. The right side of the dam continues to breach. |
| 19.25 | The breach of the right side of the dam is complete. The dam is washed out along the width of 58 m. The reservoir releases the remaining water. |
| 21.00 | The reservoir is empty. |

the German side and the city of Zgorzelec on the Polish side (the peak of the wave in Zgorzelec was at 6:40 UTC). It was the largest flood ever recorded in this area, and it caused losses on the German and Polish side of the border. The historic centers of both cities were partially flooded, and the water depth was up to 1.5 m. Supplement files Sup 1 contain pictures of the flooding, and a film is easily accesible under the web site: https://www.youtube.com/watch?v=ZCYvdTymuAw.



**Table 2.** Characteristics of the dam's failure

| HYDRAULIC CHARACTERISTICS | | | | |
|---|---|---|---|---|
| Reservoir storage to the crest level | Surface area to the crest level | Volume stored related to $H_w$ | Depth behind dam at breach inception | Peak discharge |
| $V$ | $A$ | $V_w$ | $H_w$ | $Q_p$ |
| million m$^3$ | m$^2$ | m$^3$ | m | m$^3$s$^{-1}$ |
| 8.310 | 1,900,000 | 8,541,000 | 11.70 | 1380 |

| EMBANKMENT DIMENSIONS | | | | |
|---|---|---|---|---|
| Max height[a] | $H_d$ | m | 11.6 | |
| Crest width | $W_c$ | m | 5.0 | |
| Bottom width | $W_b$ | m | 68.8 | |
| Average width | $W_{avg}$ | m | 50.0 | |
| Upstream slope | $Z$:1 | - | 0.333 | |
| Downstream slope | $Z$:1 | - | 0.4 | |

| BREACH CHARACTERISTICS | | | LEFT DAM | RIGHT DAM |
|---|---|---|---|---|
| Depth above breach (max) | $H_b$ | m | 11.6 | 11.6 |
| Average depth[b] | $H_{bavg}$ | m | 6.0 | 9.7 |
| Top width | $B_t$ | m | 106.0 | 58.0 |
| Bottom width | $B_b$ | m | 21.8 | 58.0 |
| Average width | $B_{avg}$ | m | 64.0 | 58.0 |
| Eroded volume | $V_{er}$ | m$^3$ | 25,130 | 20,860 |
| Breach formation time[c] | $T_f$ | h | 1.78 | 2.16 |
| Empty time[d] | $T_e$ | h | 3.00 | 2.75 |

[a] Difference between the lowest point of natural ground behind the dam and the dam crest level. [b] Average depth along the left and right dam axis, respectively. [c] Breach formation times provided by (Froehlich, 2008). Considered "length of time needed for the final trapezoidal breach to form, which takes place after the breach initiation phase". [d] Times from the overflow to the emptying of the reservoir.

## 2.6 Field observations

A field survey was carried out after the flood in order to collect data on the flood wave passage along the Lusatian Neisse river and its major tributaries (IMGW, 2011). A number of eye witnesses of the flood were interviewed, including several local authority representatives. The maximum water elevation marks were sought and fixed for further geodesy recording at over 50 locations, including upstream and downstream of the Niedów dam. In some cases, clear water lines were found on walls in the form of sediment and residue marks or washed out dirt, but in several locations the high water marks were only approximate,





and based on residues found on trees, bridge piers or decks. Therefore, the error of maximum water elevation may vary from a few millimeters to ca. 0.4 m. The ranges of the flooded area were also determined in the field - first marked on a map, and then digitized. The time of the flood peak passage on the Lusatian Neisse River at the section located next to the Witka River mouth, due to the fact that there was no water gauge station on this section of the river, was determined based on interviews of inhabitants. It was found that the culmination of the flood occurred between 2 and 3 a.m. on August 8, 2010, and therefore 8

hours later than the peak outflow from the failed Niedów dam. This information helps to reconstruct the flood wave hydrograph, to quantify the argued effect of coincidence flood waves from the two rivers, and to define the upper boundary conditions for the hydrodynamic model.

## 3   2D modeling of flood routing

Due to the topographical complexity of the river, including its locally meandering character and two-dimensional flow patterns,

flood routing using a two-dimensional modelling is adopted herein. This approach enables the variability of the flood wave along the Nysa Łużycka River to be best mapped in a function of time, while also taking into account the Niedów dam failure. The particular purpose of the modelling was to determine the outflow hydrogram from the reservoir, and ultimately to assess its impact on the flooding downstream.

### 3.1   Model area and boundary conditions

The 2D model domain begins with a section located several hundred meters above the mouth of the Witka River, and ends in the city of Zgorzelec that is located downsteam of the Nysa Łużycka River. The end section of the Witka River, beginning close to the dam, was modelled along with a Mill channel running through the village of Radomierzyce. The tributaries of the Pliessnitz River and the Czerwona Woda River were also included. An important aspect of the flood routing was also to restore the overflow volume through the left embankment to the Berdzorfer Lake, which is an artificial post-mining lake that received

a substantial amount of water during the flood. The considered hydrological scheme, a part of which was implemented into the 2D model, is presented in Figure 10. The unsteady state conditions were defined as follows:

$$Q_{NL,in}(t) + Q_{ND}(t) + Q_P(t) + Q_{CW}(t) - Q_B(t) = Q_{NL,Z}(t) \tag{1}$$

where: $Q_{NL,in}(t)$ – the discharge hydrograph for the Lusatian Neisse (considered as the upper boundary condition) - preliminarily interpolated from the two neighbouring gauge stations (uncertain, to be verified); $Q_{ND}(t)$ – the hydrograph of the

outflow through the Niedów dam (unknown); $Q_P(t)$, $Q_{CW}(t)$ – the flow rate hydrographs for the tributaries of the Pliessnitz River and the Czerwona Woda River (known); $Q_B(t)$ – the inflow to Berzdorf lake (for which only the total volume of inflow, rangeing from 3.5 to 4 mln m$^3$, was known. This water was retained in the reservoir and did not return to the river valley); $dV(t)$ – the change in the retention volume of the river valley; $Q_{NL,Z}(t)$ – the discharge hydrograph for the Zgorzelec gauge station; the discharges were calculated based on the flow rate curve for this station.



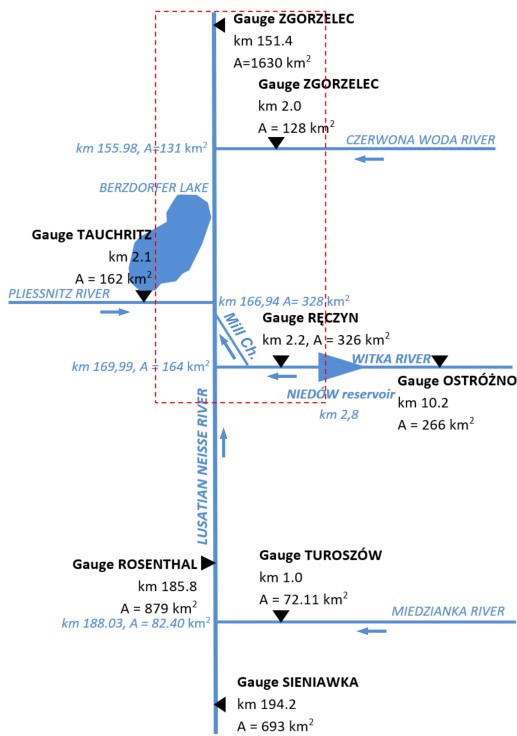

**Figure 10.** Hydrological scheme for the modelled domain (the dashed line indicates part of the 2D model)

Importantly, the measured discharge at the Zgorzelec cross-section during the maximum water level was 1040 $\mathrm{m^3s^{-1}}$. This is somewhat more than the value obtained from the extrapolated rating curve, i.e. 980 $\mathrm{m^3s^{-1}}$. Despite this difference, the downstream outflow is relatively well defined and reliable. This enables the inflows $Q_{NL,in}(t)$ and $Q_{ND}(t)$, while taking into account the additional inputs of the Pliessnitz and Czerwona Woda rivers (which were relatively insignificant), to be found. The maximum discharge rates of these tributaries were 46 and 36 $\mathrm{m^3s^{-1}}$, respectively.

**3.2    Model set up**

The two-dimensional modelling was executed using MIKE21 software (McCowan et al., 2001; DHI, 2011). This is a commonly used software by researchers and consultants for flood simulations (Kho et al., 2009; Ahmad and Simonović, 2000). The development of the model included the following works: i) preparation of the digital elevation model (DEM) based on both Polish and German data sets (selected DEM - DGM_Q1 with horizontal resolution 5 m x 5 m); ii) generation of the bathymetry

of the main river channel based on field surveyed cross-sections (every 300 m in average) by making use of a ArcGIS linear interpolation; iii) generation of the calculation bathymetry with a regular grid resolution of 5 m x 5 m by merging the main channel bathymetry with the DEM; iv) implementation of hydrotechnical structures and buildings, as well as linear structures surveyed in the field (e.g. embankments), by adjusting the ordinates of corresponding grid cells; v) preparation of a raster map




of the initial roughness on the basis of land cover maps and aerial photos; in total 15 roughness classes were distinguished –
for the main channel and open surface waters, grassland and tree areas, bushes, paved surfaces, roads, etc.; vi) formulation of
boundary conditions in the form of water level and discharge series (IMGW, 2011). The computation bathymetry and roughness
parameters (expressed by the velocity coefficient $M = 1/n$, where $n$ is the Manning coefficient) are provided as supplementary
data files (Sup 2 and Sup 3, respectively), while the roughness parameter was preliminarily determined with guidance from
Arcement and Schneider (1989) and Morvan et al. (2008). The M-values for the main channel ranged from 22 (meandering
sections) to 36 $\mathrm{m}^{1/3}\mathrm{s}^{-1}$ (regular channel in Zgorzelec). Since the Lusatian Neisse River was modeled as an open-ended reach,
the downstream boundary condition was set as a water elevation in a function of time. This was directly adopted from the
observations of the Zgorzelec gauge station, because the modelled area ends on the cross-section of this station. The size of the
modelled area was 13.3 km by 5.0 km, the total number of grid cells was 2.65 million, and the computation step was from 0.5
to 0.75 seconds. The assumed computation step of the calculations was meant to limit the Courant number to the value of 1.0
for the purpose of obtaining: numerical stability, the accuracy of the calculations, and a computation time of the simulations
that is not too long.

## 4   Results

The flooding along the Lusatian Neisse in the studied case is a combination of two major flood waves that originated from the
upstream river section and the Niedów reservoir outflow. The flood hydrographs constitute the two upper boundary conditions
of the model (see Fig. 11), and they are reconstructed.

     The reconstruction of these hydrographs was iterative, and relied on a number of computations that were executed with
adjusted shapes of inflow hydrographs in order to satisfy Eq. 1. The adjustments mainly concerned the value of the wave peak
and, to a lesser extent, the time of the wave peak, which was determined on the basis of the testimony of witnesses. In addition
to this, the analysis of several different time failure scenarios led to the conclusion that limited variations in the dynamics of the
failure have a negligible practical influence on the spatial envelope of the maximum values of the water depth and downstream
discharge. This is in accordance with the research results of Pilotti et al. (2011). Moreover, a simultaneous sensitivity analysis
was performed to identify and analyze the influence of the change of the roughness Manning coefficient on the peak flow and
wave front propagation downstream. The roughness parameter is considered to be the most influential for the flow, as was the
case in other numerical studies (Hall, 2005; Pappenberger et al., 2005). Therefore, the roughness values were changed in order
to bring the calculated hydrograph of the flow in Zgorzelec in line with the hydrograph measured on the water gauge, as well
as to obtain a good agreement of the calculated water table with the measured marks of the high water.

     As a result of the conducted simulation, Figure 12 illustrates the flooding at 10:00 on August 8, 2010, when the flood
peak reached the city of Zgorzelec. This figure also indicates the location of the high water marks (denoted as WW) on the
right bank. The final computation is considered as a satisfactory reconstruction of the flood in terms of water level, flooding
extent, discharge rate, timing, and water volume. The calculated ordinates of the water table correspond relatively well with
the high water marks that were measured on site after the flood. The differences between the calculated values and those that

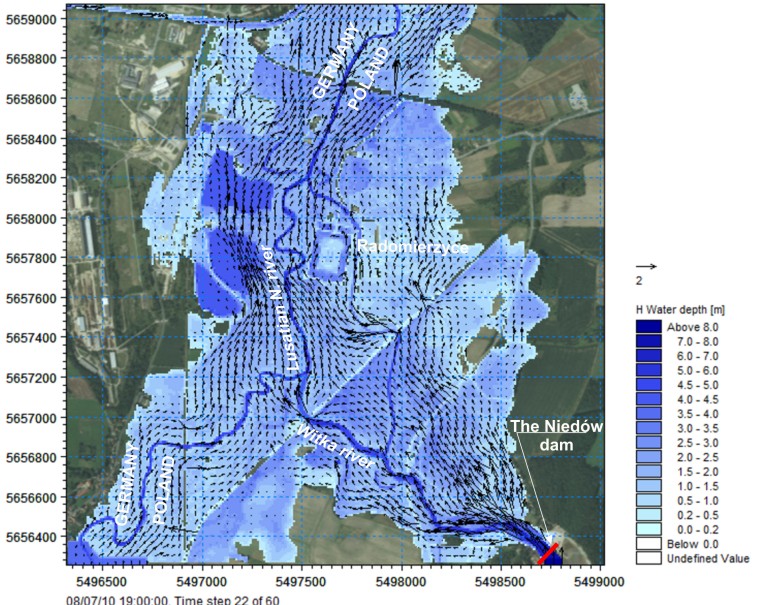

**Figure 11.** Simulated water depth and flow velocity vectors downstream from the Niedów dam [Polish UWPP 2000 Reference system (m)]

were determined vary from a few centimeters to approx. 0.3 m (Tab. 3). In addition, the calculated overflow volume to the Berzdorfer Lake amounted to 3,783 million $m^3$, which is in accordance with the data provided by the German party. This value was assessed based on the water level increase in the lake before and after the flood passage. The resulting upstream
dam breach hydrograph $Q_{ND}(t)$ was determined for a peak discharge of 1380 $m^3s^{-1}$ at 18:20 (see Fig. 13). The total volume of released water due to the dam's failure was equal to 22 million $m^3$, which is about 5 mln $m^3$ more than the inflow to the reservoir.

Finally, Figure 14 presents the crucial discharge hydrographs, which reflect the flood wave transformation along the modelled river section. The influence of valley retention on the flood propagation is remarkable. This retention was of about 20 million
$m^3$, not counting the inflow to Lake Berzdorfer. Therefore, there was a significant reduction of the flood peak discharge from 1730 $m^3s^{-1}$ at the cross-section near the Witka mouth (km 169.5) to 950 $m^3s^{-1}$ at the Zgorzelec gauge station (km 151.4), which prevented more severe damage in the city of Zgorzelec. In addition, the discharge hydrograph at Zgorzelec shows two peaks - the first one was caused by the Niedów dam's break, and the second one was caused by the flood wave on the Lusatian Neisse, which culminated, importantly and fortunately, about eight hours later than that of the Witka River. As depicted in
Figure 14, the travel time of the first flood peak from the outflow from the Niedów reservoir to the Zgorzelec gauge station took about seven hours, while the second peak of the Lusatian Neisse traveled for about 4.5 hours. This is reasonable, taking into account the fact that the second peak travelled over areas that were already flooded, and therefore it was faster than the first one.



**Table 3.** Comparison between the measured and calculated water levels (for location see Fig. 12)

| Water marks | H measured (m) | H calculated (m) | Difference (m) |
|---|---|---|---|
| WW1 | 200.73 | 200.656 | -0.074 |
| WW2 | 199.04 | 199.121 | 0.081 |
| WW3 | 199.63 | 199.539 | -0.091 |
| WW4 | 197.72 | 197.618 | -0.102 |
| WW5 | 198.03 | 197.470 | -0.56 |
| WW6 | 197.66 | 197.382 | -0.278 |
| WW7 | 197.56 | 197.383 | -0.177 |
| WW8 | 197.22 | 197.327 | 0.107 |
| WW9 | 195.17 | 195.09 | -0.08 |
| WW10 | 193.91 | 193.942 | 0.032 |
| WW11 | 191.53 | 191.312 | -0.218 |
| WW12 | 191.36 | 191.125 | -0.235 |
| WW13 | 190.22 | 190.008 | -0.212 |
| WW14 | 189.09 | 189.203 | 0.113 |
| WW15 | 188.31 | 188.466 | 0.156 |
| WW16 | 188.53 | 188.463 | -0.067 |
| WW17 | 188.77 | 188.727 | -0.043 |
| WW18 | 185.43 | 185.498 | 0.068 |
| WW19 | 184.79 | 184.658 | -0.132 |
| WW20 | 182.77 | 182.754 | -0.0 |

## 5  Conclusions

The literature review and the current case study demonstrate that the dam breach mechanism and its prediction is an extensively studied and complex subject. There is a variety of failure modes and possible approaches to quantitatively assess the dynamics and consequences of the dam break. Thus, the study aimed to reconstruct and explain the catastrophic event of the Niedów dam failure in order to contribute to the current database concerning the developments of dam breaches. As a result, a detailed description of the dam breaching mechanism is provided with the final breach parameters, which can be used for statistical

analyses or for the development of a model that is based on the description of the physics of this phenomenon. A particular feature of the Niedów dam was the fact that the homogenous embankments made of sand and gravel had a concrete facing, which acted as an impermeable barrier. This, along with the asphalt road on top, substantially affected the process of the washing out of both sides of the embankments, which was remarkably different and longer than what can be expected in the case for homogenous earth embankments. The slower washing away of the dam resulted in a lower peak value of the wave
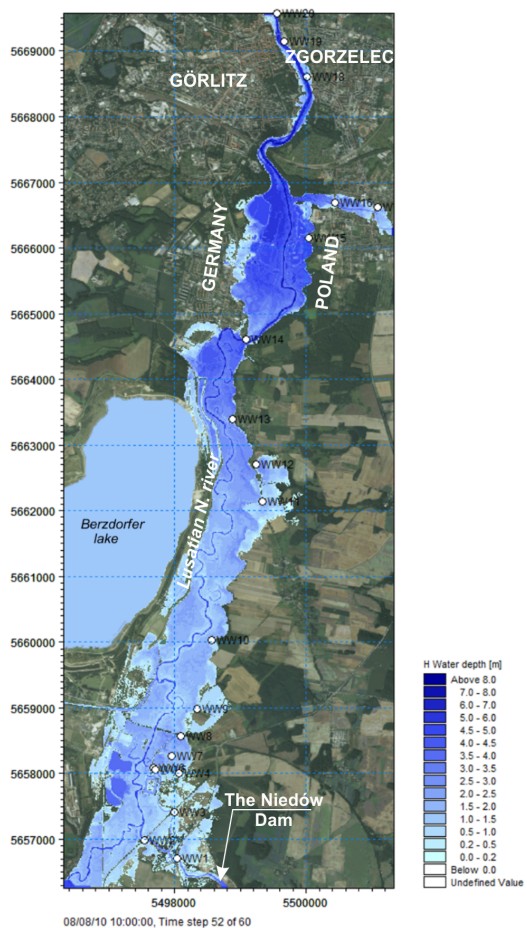

**Figure 12.** 2D simulation of the flood on the Lusatian Neisse in 2010 (WW - high water mark, whereas WW20 indicates the gauge station Zgorzelec)

flowing out of the breach. Therefore, according to the authors of the article, when preparing a forecast of a concrete faced dam failure, this fact should be taken into account in the analysis of the flood risk, as it may significantly affect the assessment of the consequences of the disaster. It can also be concluded that the concrete facing of dams can be a measure for limiting the peak outflow of a potential dam breach and for reducing the risk of flooding.

      Another particularity of this case study is that the catastrophic flooding along the Lausatian Neisse was a superposition of

two floods, with the consequences of both needing to be explained. The paper therefore presents an implementation of a 2D hydrodynamic model for simulating flood wave propagation along the Lusatian Neisse River, while at the same time taking into account the Niedów dam failure. With the use of this model, the unknown upper boundary hydrographs of this complex flood situation, in particular the outflow from the reservoir, are determined in an iterative way, while also making use of the mass conservation principle in unsteady state simulations. This modelling approach is considered to be an alternative to the


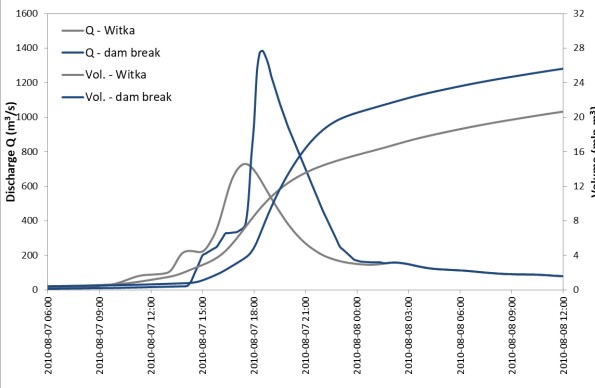

**Figure 13.** Discharge hydrographs for the inflow to the reservoir and the outflow from the reservoir. The later is the result of the dam's failure determined during 2D calculations. The vertical axis on the right shows the cumulated curve of the outflow from the reservoir - the final difference between the curves of both scenarios indicates the water retention in the reservoir under normal conditions.

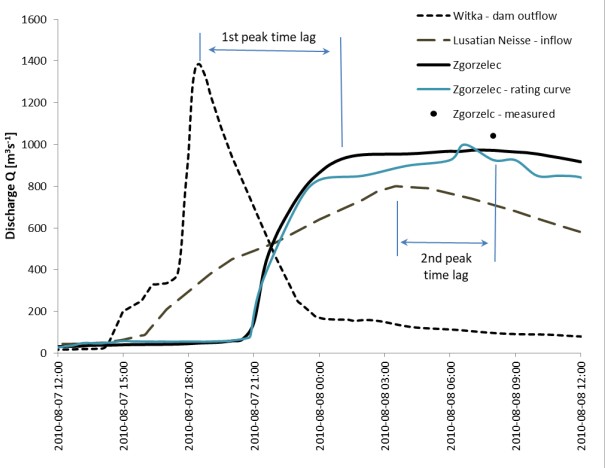

**Figure 14.** Hydrographs of discharges of the Witka and the Lusatian Neisse River

assessment of outflow hydrographs based on statistical formulas, which are not successfully applied because of the present complexity of the breaching process. Remarkably, as a result of the executed hydraulic modelling in a data limited situation, relevant answers concerning flood related damage to various stakeholders in a bilateral, cross-border context could also be provided.

*Author contributions.*  SK and RB collected the data, conceived the study, interpreted the results and wrote the paper. SK assessed the dam
break formula performance while RB conducted the flood routing. The authors revised and approved the paper.



*Competing interests.* The authors declare that they have no conflict of interest.



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
