# Peer review of "The catastrophe of the Niedów dam - the causes of the dam's breach, its development and consequences"

_Natural Hazards and Earth System Sciences, 2021_

## Referee Comment (RC1)

This paper documents an important flood event that was caused (as the reader finds out by himself step by step reading the paper and then explicitly finds explained at line 243) by the superposition of two floods, one of which was caused by the Niedów dam-breach.

I reviewed the first version of this paper at the beginning of the year and its readibility has been definitely improved but however I am sorry to come to the conclusion that I still believe that it is unsuitable for publication in its current form. Apart from a set of typos, naïve statements (e.g., in the Abstract, "The flood event occurred downstream from the dam "), uncorrect use of technical terms (e.g.,  water table, that is a term used in groundwater terminology, in place of water surface; velocity coefficient for Strickler's coefficient) and undocumented statements, there is a fundamental bias that has not been solved yet.

At the core of the simulation and of all the reasonings there is the use of equation 1 (that is still written in a wrong way) to compute the outflow hydrograph from the Niedov dam. The point is that, even disregarding the time distribution of the outflow to the Berzdorf lake (only the overall volume spilled in the lake is documented in the paper but not its time distribution) and the variation of the stored volume in the floodplain (that does not appear in eq. 1 – and that is the reason for which eq. 1 is wrong -but that that must be calculated by MIKE21) , there are two unknowns functions in the equation: the discharge hydrograph QND from the Niedov dam and the discharge hydrograph from the Lusatian river QNL: this is explicitly said: "This enables the inflows QNL,in(t) and QND(t), …, to be found".

With a single constraint (equation 1) there are an infinite possibility to find different sets of QNL,in(t) and QND(t) to match QNL,Z (t), i.e., the discharge hydrograph for the Zgorzelec gauge station.

It si true that a contradictory and mysterious phrase at line 208 writes: "QNL,in (t), …preliminarily interpolated from the two neighbouring gauge stations (uncertain, to be verified)" but this piece of information, if present (what is the meaning of uncertain, to be verified), does not show up in any other part of the paper.

Accordingly, failing to detail this fundamental point, as well other informations partly listed in the following, in my opinion the colored maps of Fig. 11 and 12, have no particular relevance because the overall procedure looks flawed.

Follow a list of more particular but important details that show that the paper has not yet been carefully reviewed by the Authors

| Line | text | Observation |
|------|------|-------------|
| 16 | cauese | cause |
| 82 | Maximum yield of the weir | ? |
| 118 | The return period of the flood.. | On the basis of what ? Analysis of Rainfall, maximum discharge ? Measured where ? |
| 120 | On the 7th of August at the Ostrózno gauge station, the highest ˙ water level of the flash flood occurred at 16:40. The R̦eczyn gauge station was recording the water level until the time of 15:20, and thus until it was destroyed due t | Here, as in many following points, you mention to the existence of gauge stations, but without showing the available data. A graph should be added with all the available measured level or discharge hydrographs at the relevant stations during the flood. On the some graph the timing of the most important events listed |
| 129 | On the 7th of August, the estimated | Delete. Already said at line above |

| | | |
|---|---|---|
| | flood rate was 615 m3s/1 | |
| 134 | The water level … | Is there any recording of the water level as a function of time ? It would be important to show the elevation as a function of time and in correspondence the operation of the gates |
| 138 | After the water level exceeded the edge of the repaired gate, | What do you mean ? Explain better |
| 142 | which is documented in Sup. 1 | No, in the supplementary materials there are some pictures (where ?) and two maps. No other material is available on the dam breach |
| 167 | Radomierzyce through the Mill channel. | Every place that is mentione in the paper must be retracebale on the map. I don't see this place neither in Figure 1 nor 6 which are the ones mentioned so far in the paper. At the same time, regarding the name of the rivers, you must use always the same name (Nysa Łuzycka River and e Lusatian Neisse River are probably the same river) and it must be the one that appears on the map |
| 170 | destruction (disintegration) of the buildings | Do you mean collapse ? |
| 173 | it flooded the Hagenwerder estate | As at line 167 |
| 175 | city of Zgorzelec on the Polish side (the peak of the wave in Zgorzelec was at 6:40 UTC) | Here one starts realising that a second flood is superimposed to the dam breach wave but considering that you do not clearly explain this point in advance one is left puzzled at how it is possible that the dam breach wave takes so long to get to this town. |
| 203 | To restore | ? |
| 207 | Equation 1 | This equation is wrong because it does not include the dV/dt term. In the following text you list dV, that does not appear in the equation but this is another error because dV is a volume and is not dimensionally coherent with discharge Q. |
| 213 | dV | dV does not appear in the equation but this is another error because dV is a volume and is not dimensionally coherent with discharge Q. |
| 215 | Measured discharge | Did somebody actually measure the discharge during the flood ? This is a complex task: how did they do it ? |
| 218 | This enables the inflows QNL,in(t) and QND(t), while taking into account the additional inputs of the Pliessnitz and Czerwona Woda rivers (which were relatively | In my opinion there are a lot of ways to match the measured discharge with different input hydrographs. You do not discuss this point in sufficient detail. |

| | insignificant), to be found. | |
|---|---|---|
| 231 | Velocity coefficient 1/n | This is what everybody call Strickler's coefficient. By the way in the map in the supplementary file you show Strickler's coeffciect as low as below 2.5. This is actually an unbelievable value: which type of ground cover did you model with this low value ? |
| 245 | Fig 11 | Why ? |
| 258 | the flooding at 10:00 on August 8, 2010, when the flood peak reached the city of Zgorzelec | From figure 12 one would say between 6 and 9 AM |
| 264 | based on the water level increase in the lake | Having the variation as a function of time would be another important calibration point. But nothing is shown in the paper about this important point |
| 266 | The total volume of released water due to the dam's failure was equal to 22 million m3 | No, this is false. Due to the dam failure only the volume stored in the reservoir was released, |
| 275 | the travel time of the first flood peak from the outflow from the Niedów reservoir to the Zgorzelec gauge station took about seven hours | This is really strange, considering that the two cross section are probably 10 kms apart. It would imply an average velocity of about 0.4 m/s that is really low for a dam breach flood. This point should be discussed better… |
| 285 | A particular feature of the Niedów dam ….

was the fact that the homogenous embankments made of sand and gravel had a concrete facing, which acted as an impermeable barrier. | To be honest I was surprised to hear than an earth dam was totally made with sand with a permeability coefficient of $2.8 \times 10$-3 ms−1, that is huge, without any impermeable core. Accordingly, the waterproof coating on the inner side of the embankments was totally mandatory and is certainly not a "particular feature" but a must. Rather, I would have concentrated my discussion on two considerations: 1) the 1/100 year return time for the design discharge of the dam was clearly inadequate. 2) the maintanence of the hydropower station that apparently led to the cut-off of the power supply and so contributed to the disaster, was scheduled without the needed attention to the possible occurrence of a flood in that period of the year. |

| Table 1 | Apparently the dynamic of the gates opening is in contraddiction with the text where you write that "The crew still tried to open more gates manually from the dam's crest, but were unsuccessful." Accordingly, one would expect that after |
|---|---|

| | |
|---|---|
| | 15:36 the gates stay fixed in their position.
Moreover, if a leve recording is available at Ostrozno it should be plotted as a function of tme |
| Table 2 | The peak discharge is a result of your model ? You must specify it |
| Figure 2 | Add ruler for distances |
| Figure 6 | You show the state borders (which are pretty unrelevant and should be dropped) but not the border of the catchments |
| Figure 10 | Gauge ZgorZelec appears twice. Which is the right one ?
Moreover in the paper all the level/discharge recording at the different gauge stations must be shown as a function of time during the event. |

---

## Author Comment (AC1)

Reviewer comment:

This paper documents an important flood event that was caused (as the reader finds out by himself step by step reading the paper and then explicitly finds explained at line 243) by the superposition of two floods, one of which was caused by the Niedów dam-breach.

I reviewed the first version of this paper at the beginning of the year and its readibility has been definitely improved but however I am sorry to come to the conclusion that I still believe that it is unsuitable for publication in its current form. Apart from a set of typos, naïve statements (e.g., in the Abstract, "The flood event occurred downstream from the dam "), uncorrect use of technical terms (e.g., water table, that is a term used in groundwater terminology, in place of water surface; velocity coefficient for Strickler's coefficient) and undocumented statements, there is a fundamental bias that has not been solved yet.

At the core of the simulation and of all the reasonings there is the use of equation 1 (that is still written in a wrong way) to compute the outflow hydrograph from the Niedov dam. The point is that, even disregarding the time distribution of the outflow to the Berzdorf lake (only the overall volume spilled in the lake is documented in the paper but not its time distribution) and the variation of the stored volume in the floodplain (that does not appear in eq. 1 – and that is the reason for which eq. 1 is wrong -but that that must be calculated by MIKE21) , there are two unknowns functions in the equation: the discharge hydrograph QND from the Niedov dam and the discharge hydrograph from the Lusatian river QNL: this is explicitly said: "This enables the inflows QNL,in(t) and QND(t), …, to be found".

With a single constraint (equation 1) there are an infinite possibility to find different sets of QNL,in(t) and QND(t) to match QNL,Z (t), i.e., the discharge hydrograph for the Zgorzelec gauge station.

Authors answer

The numerical modelling problem is generally formulated in Eq. 1. Initially the term for retention dV(t) was not included as it is an inherent part of the 2D model solution. Unfortunately, corrected Eq. 1 went wrong in the final revision/edition stage followed by an oversight, but the term dV(t) is included in the text below.

Also the overflow to the Berzdorfer lake is calculated by the program. This flow over the embankment can be extracted in a function of time, however, there are no field data to verify this variation. In addition, it would rather have a relative minor impact on the flood propagation (taking into account the total valley retention of ca. 20 million m3), so the total overflow volume match is regarded mandatory.

Indeed, the reviewer's concern about the solution of Eq. 1 is right, given that there are two unknowns in one equation. One can further suggest an infinite number of solutions. However, there is much less possibilities given the restrictions and conditions the model had to satisfy (cf. section: *2.6 Field observations*). There are known upper inflow peaks timing and high water level marks and finally the outflow hydrograph (Zgorzelec cross section). Using these one can search for approximate hydrogram shapes. Of course, there is still a possibility of a variety of peak flows and hydrograph shapes, but those found in the paper reasonably satisfied the observation in the limit of the afforded 2D modelling. An iterative approach (trial and error) is applied as no other choice here, and the authors consider this approach as a major achievement of the work. We also see now the need for more clarification in this respect in the text and we are indeed thankful to the Reviewer for his comments. Obviously, the task was complicated by the data limited situation as it is typical for such events. Nevertheless, for the sake of this work the authors collected a vast amount of data which was also a great effort of different teams. Yet, these results have already been discussed multiple times and accepted by German parties involved in the International Commission for the Protection of the Odra River against Pollution.

Accordingly, failing to detail this fundamental point, as well other informations partly listed in the following, in my opinion the colored maps of Fig. 11 and 12, have no particular relevance because the overall procedure looks flawed.

We hope that after these fundamental explanations the Reviewer will change his opinion on the numerical approach and modelling results. Further, there are more detailed clarifications following the Reviewer's work:

Follow a list of more particular but important details that show that the paper has not yet been carefully reviewed by the Authors

| Line | text | Observation |
|------|------|-------------|
| 16 | cauese | cause
corrected |
| 82 | Maximum yield of the weir | ? |
| 118 | The return period of the flood.. | On the basis of what ? Analysis of Rainfall, maximum discharge ? Measured where ?
The return period is evaluation based on the statistics of the yearly peak flows for the gauge stations. Explanation and reference added to the text. |
| 120 | On the 7th of August at the Ostrózno gauge station, the highest ˙
water level of the flash flood occurred at 16:40. The Ręczyn gauge station was recording the water level until the time of 15:20, and thus until it was destroyed due to | Here, as in many following points, you mention to the existence of gauge stations, but without showing the available data.
A graph should be added with all the available measured level or discharge hydrographs at the relevant stations during the flood. On the some graph the timing of the most important events listed.
A graph presenting recorded water levels will be added. |
| 129 | On the 7th of August, the estimated flood rate was 615 m3s/1 | Delete. Already said at line above
Done |
| 134 | The water level ... | Is there any recording of the water level as a function of time ? It would be important to show the elevation as a function of time and in correspondence the operation of the gates.
There are data for all three closures available until the power supply was working. We reconsider adding a graph to the text. |
| 138 | After the water level exceeded the edge of the repaired gate, | What do you mean ? Explain better
This sentence was changed to:
After the water level exceeded the edge of the maintenance gate at the inlet to hydropower channel. |

| 142 | which is documented in Sup. 1 | No, in the supplementary materials there are some pictures (where ?) and two maps. No other material is available on the dam Breach
Finally not added due to copyright issues. Corrected. |
|---|---|---|
| 167 | Radomierzyce through the Mill channel. | Every place that is mentione in the paper must be retracebale on the map. I don't see this place neither in Figure 1 nor 6 which are the ones mentioned so far in the paper. At the same time, regarding the name of the rivers, you must use always the same name (Nysa Łuzycka River and e Lusatian Neisse River are probably the same river) and it must be the one that appears on the map
The Mill channel is not indicated on Figure 1 and 6 due to the scale of the map (for clearance). Instead the Mill channel is indicated on Fig. 10, with a reference in text. |
| 170 | destruction (disintegration) of the buildings | Do you mean collapse ?
Yes. corrected |
| 173 | it flooded the Hagenwerder estate | As at line 167
The Hagenwerder estate is located next to the Pliessnitz mounth. |
| 175 | city of Zgorzelec on the Polish side (the peak of the wave in Zgorzelec was at 6:40 UTC) | Here one starts realising that a second flood is superimposed to the dam breach wave but considering that you do not clearly explain this point in advance one is left puzzled at how it is possible that the dam breach wave takes so long to get to this town.
Explanation amended in the text. |
| 203 | To restore | Such a wording was found in a publication.n
We change it e.g. 'to determine' (we noted such a wording in a publication). |
| 207 | Equation 1 | This equation is wrong because it does not include the dV/dt term. In the following text you list dV, that does not appear in the equation but this is another error because dV is a volume and is not dimensionally coherent with discharge Q.
The equation is corrected, the following text included the term dV (t) |
| 213 | dV | dV does not appear in the equation but this is another error because dV is a volume and is not dimensionally coherent with discharge Q.
as above |

| 215 | Measured discharge | Did somebody actually measure the discharge during the flood ? This is a complex task: how did they do it ?
The discharge was measured by a team of the hydrometry service of the Institute of Meteorology and Water Management (measurement made from the bridge in Zgorzelec). |
|-----|--------------------|-------------------------------------------|
| 218 | This enables the inflows QNL,in(t) and QND(t), while taking into account the additional inputs of the Pliessnitz and Czerwona Woda rivers (which were relatively insignificant), to be found. | In my opinion there are a lot of ways to match the measured discharge with different input hydrographs. You do not discuss this point in sufficient detail.
Using an iterative approach, by trial and error one can determine approximate input hydrographs given the peak timing and additional data available / restrictions. Additional explanations on this procedure to be added in the corrected text. |
| 231 | Velocity coefficient 1/n | This is what everybody call Strickler's coefficient. By the way in the map in the supplementary file you show Strickler's coeffciect as low as below 2.5. This is actually an unbelievable value: which type of ground cover did you model with this low value ?
The M coefficient (MIKE notation) or Stickler coefficient is sometimes called as the velocity coefficient because the flow velocity is proportional to it. As we keep M notation we further assume the term 'roughness parameter (l.232)". By the way – for building areas the M value was set to 1. |
| 245 | Fig 11 | Why ?
(see Fig. 11) - removed. |
| 258 | the flooding at 10:00 on August 8, 2010, when the flood peak reached the city of Zgorzelec | From figure 12 one would say between 6 and 9 AM.
As a result of the conducted simulation, Figure 12 illustrates the flooding at 10:00 on August 8, 2010, soon after the flood peak reached the city of Zgorzelec. |
| 264 | based on the water level increase in the lake | Having the variation as a function of time would be another important calibration point. But nothing is shown in the paper about this important point
There are no data regarding the variation of the overflow to the Berzdorfer; only the total volume is known. Hence, the overflow volume computed by the program was linked with the local water level, which |

| | | depended on the maximum discharge in the river (including the outflow from the broken Niedów dam) and to the roughness values. |
|---|---|---|
| 266 | The total volume of released water due to the dam's failure was equal to 22 million m3 | No, this is false. Due to the dam failure only the volume stored in the reservoir was released, Right, additional outflow from the reservoir was ca. 5 million m3. |
| 275 | the travel time of the first flood peak from the outflow from the Niedów reservoir to the Zgorzelec gauge station took about seven hours | This is really strange, considering that the two cross section are probably 10 kms apart. It would imply an average velocity of about 0.4 m/s that is really low for a dam breach flood. This point should be discussed better… This is not strange given the topography, relatively low slope, meandering river character, vegetation influence, retention increased by two weirs. The model simulations are in agreement with the observation at the Zgorzelec gauge station. This issue will be additionally commented. |
| 285 | A particular feature of the Niedów dam …. was the fact that the homogenous embankments made of sand and gravel had a concrete facing, which acted as an impermeable barrier. | To be honest I was surprised to hear than an earth dam was totally made with sand with a permeability coefficient of 2.8×10-3 ms−1, that is huge, without any impermeable core. Accordingly, the waterproof coating on the inner side of the embankments was totally mandatory and is certainly not a "particular feature" but a must. Rather, I would have concentrated my discussion on two considerations: 1) the 1/100 year return time for the design discharge of the dam was clearly inadequate. 2) the maintenance of the hydropower station that apparently led to the cut-off of the power supply and so contributed to the disaster, was scheduled without the needed attention to the possible occurrence of a flood in that period of the year. Description corrected |

| Table 1 | Apparently the dynamic of the gates opening is in contradiction with the text where you write that "The crew still tried to open more gates manually from the dam's crest, but were unsuccessful." Accordingly, one would expect that after |
|---|---|

| | 15:36 the gates stay fixed in their position. |
|---|---|
| | Moreover, if a leve recording is available at Ostrozno it should be plotted as a function of time |
| | To be clarified |
| Table 2 | The peak discharge is a result of your model ? You must specify it |
| | Yes, this value is calculated by the model and will be specified accordingly. |
| Figure 2 | Add ruler for distances |
| | The ruler has been added |
| Figure 6 | You show the state borders (which are pretty unrelevant and should be dropped) but not the border of the catchments |
| | The state borders are removed |
| Figure 10 | Gauge ZgorZelec appears twice. Which is the right one ? |
| | Moreover in the paper all the level/discharge recording at the different gauge stations must be shown as a function of time during the event. |
| | A figure showing water level hydrographs for gauging stations will be added. |

---

## Author Comment (AC2)

Reviewer comments:

The problem is very interesting, but its discussion is sadly incomplete. Analysing the current version of the article, clear conclusions for the ICOLD cannot be drawn, and thus the acceptance of the submitted description for publication in an international journal is problematic.

I wish to analyse a diagram that is crucial for a discussion of disasters affecting hydraulic structures, such as dams. Key details for an analysis include:

Functions to be performed by the structure – a description Geomorphological and hydrological conditions Design guidelines (applicable during design work), data adopted for designing purposes, obtained final flow capacity parameters of the structure, geotechnical parameters of the structure, device output curves A short operational description of the structure, technical assessments made, hydrological events, structure condition (maintenance status), changes in geotechnical parameters, dislocation of land-surveying points, filtration through the structure and results of control operations complete probabilistic and physical characteristics of the input function that directly caused the disaster indirect conditions, here e.g. instructions for water management in the reservoir as a principal document binding upon the operator and deviations in control processes with their reasons An analysis of simulation results and an assessment of potential differences compared to ICOLD data, applicable assessment methods that were used (e.g. empirical formulae) If a structure with the same crosssection is to be reconstructed, a rationale must be given with applicable regulations and new characteristics of devices

Authors answer:
The authors thank the Reviewer for his comments. First of all we want to answer that the purpose of our paper was not preparing the data for an ICOLD data base. Satisfying this expectation the paper would have a substantially larger number of pages, probably beyond the editorial acceptance. We tried to distill the most relevant data and information, even though the paper became already sizeable, going to increase after reviewer comments.

The items indicated above are not explained in the article (items 3, 5, 7, 8) or are incompletely explained (all remaining items). The title indicates that the article was aimed to describe the causes of disaster, its development and consequences. All those elements can be identified but cannot be characterized as scientific. The article is structured as a superficial report on a failure, without any scientific commentary and references to formulas that are currently used to assess and analyse disasters (an attempt to analyse the problem scientifically was made in the previous version). The manuscript lacks a scientific commentary substantiated by calculations.

A really tough statement. The in previous paper version presented calculations of the dam breach characteristics (breach dimensions, time, outflow) using empirical formulas gave significantly different results. Reviewer # 1 in his first evaluation stipulated that using empirical formulas for such a complex case is doubtful and prone to discussion. The authors after making this difficult exercise agreed to this comment and removed this part from the paper. Further, we would like to emphasize that the major goal

of the paper is to present the numerical modelling approach to determine the flood hydrographs on the Witka river and the Lusatian Neisse river leading to the catastrophe of the Niedów dam and material losses downstream. The complexity of this analysis is linked to the data limited situation, including the damage of the gauge stations both in the Czech Republic, Poland and Germany. Therefore the authors applied a numerical hydrodynamic model and a trial and error approach to find the unknown hydrographs. This approach in this complex situation of two superimposed floods, is the major achievement of the research, relying also on a vast number of data collected from different sources. No other choice is possible to attempt the evaluation of the outflow hydrogram. The documentation value of this work should not also be disregarded.

The proposed formula (1) ignores the physics of the phenomenon and is erroneous.

Thank you for this remark. The formal error made in the equation (already corrected) has no influence on the modelling results as the retention dynamics is included in the applied 2D model solution.

Additionally, the concept of iteration is introduced without a precise equation / system of equations explaining that concept.

As above, this procedure will also receive additional comments.

One of the most important tasks in analysing this type of disasters is to compare inflows with throughput capabilities of the structure (a capacity curve of discharge and spill devices - here omitted).

The capacity curve for this type of structure is relatively well known. Therefore, we provided only the maximum capacity at a given water level in the reservoir.

In the description of hydrological background (precipitation and flow rates), hyetograph information is omitted, and there is no reference to the probability of maximum annual flow rates being exceeded.

There is a number of meteorological station in the catchments of the Lusatian Neisse and the Witka river. The authors will reconsider including this piece of information as representative for the event under discussion. On the other hand, the purpose of the paper was not to analyse the run-off from the catchment and the exact meteorological conditioning.

Indeed, there is only a short information regarding the flood exceedance probability, i.e. the return period ranging from 100 to 200 year. A reference to this statistical evaluation can be corrected (actually it is already present as IMGW et al., 2010 – a trilateral report).

The discharge and spill devices in the structure were designed for a 1,000-year flood (estimated in the 1960s at about 650 m3 /s). There was no applicable regulation then in force, other guidelines were followed, namely Soviet standards). The guidelines in force at present (Journal of Laws Dz.U. 2007 no. 86 item 579) require that the structure be designed with parameters meeting the requirements for Class 1: a 5,000-year flood (Wpływ stanu technicznego na katastrofę zapory Niedów, Kostecki S., Rędowicz W., Machajski J., Politechnika Wrocławska, Przegląd Budowlany – 2012) – but is designed for a 1,000-year flood. There is no information about this aspect, and no comment explaining the reason for a reduced class of that hydraulic structure.

The paper may be called not to tackle a number of aspects regarding the Niedów dam in terms of construction and exploitation. Again, the paper would become sizable doing so. These issues were intentionally shortened or omitted under our statement of the scientific goal.

The key cross-sections, referring to Figure 10, do not contain flow hydrographs.

The flow hydrographs are presented in fig 13.

Table 2 contains a surprising example of consistency between calculation results and measurements, unattainable in bivariate modelling. This requires comments, especially considering that the measurements were not taken during the process but after some time.

As above stated additional comment will be included. But please note that the peak discharge in Table 2 results from the hydrodynamic modelling (will be clarified in the text).

The article is unsuitable for publication in its current version, it has to be supplemented and thoroughly restructured. Its technical language must be corrected.

A general statement. Please be more specific or provide examples. Any exact improvement suggestions will be highly appreciated.

---

## Author Comment (AC3)

Thank you very much for referee's many valuable comments and suggestions. We reply herein with the order of their appearance.

Reviewer comment:

This paper documents an important flood event that was caused (as the reader finds out by himself step by step reading the paper and then explicitly finds explained at line 243) by the superposition of two floods, one of which was caused by the Niedów dam-breach.

I reviewed the first version of this paper at the beginning of the year and its readibility has been definitely improved but however I am sorry to come to the conclusion that I still believe that it is unsuitable for publication in its current form. Apart from a set of typos, naïve statements (e.g., in the Abstract, "The flood event occurred downstream from the dam "), uncorrect use of technical terms (e.g., water table, that is a term used in groundwater terminology, in place of water surface; velocity coefficient for Strickler's coefficient) and undocumented statements, there is a fundamental bias that has not been solved yet.

At the core of the simulation and of all the reasonings there is the use of equation 1 (that is still written in a wrong way) to compute the outflow hydrograph from the Niedov dam. The point is that, even disregarding the time distribution of the outflow to the Berzdorf lake (only the overall volume spilled in the lake is documented in the paper but not its time distribution) and the variation of the stored volume in the floodplain (that does not appear in eq. 1 – and that is the reason for which eq. 1 is wrong -but that that must be calculated by MIKE21), there are two unknowns functions in the equation: the discharge hydrograph $Q_{ND}$ from the Niedov dam and the discharge hydrograph from the Lusatian river $Q_{NL}$: this is explicitly said: "This enables the inflows $Q_{NL,in}(t)$ and $Q_{ND}(t)$, …, to be found".

With a single constraint (equation 1) there are an infinite possibility to find different sets of $Q_{NL,in}(t)$ and $Q_{ND}(t)$ to match $Q_{NL,Z}(t)$, i.e., the discharge hydrograph for the Zgorzelec gauge station.

Thank you very much for commenting this equation (1). The numerical modelling problem is generally formulated in Eq. 1. Initially the term for retention $dV(t)$ was not included as it is an inherent part of the 2D model solution. Unfortunately, corrected Eq. 1 went wrong in the final revision/edition stage followed by an oversight, but the term $dV(t)$ is included in the text below.

Also the overflow to the Berzdorfer lake is calculated by the program. This flow over the embankment can be extracted in a function of time, however, there are no field data to verify this variation. In addition, it would rather have a relative minor impact on the flood propagation (taking into account the total valley retention of ca. 20 million m3), so the total overflow volume match is regarded mandatory.

Indeed, the reviewer's concern about the solution of Eq. 1 is right, given that there are two unknowns in one equation. However, there are conditions and restrictions the model had to satisfy (cf. section: *2.6 Field observations*). There are known upper inflow peaks timing and high water level marks and finally the outflow hydrograph (Zgorzelec cross section). Using these one can search for approximate hydrogram shapes. Of course, there is still a possibility of a variety of peak flows and hydrograph shapes, but those found in the paper reasonably satisfied the observation in the limit of the afforded 2D modelling. An iterative approach (trial and error) is applied as no other choice here, and the authors consider this approach as a major achievement of the work. We also see now the need for more clarification in this respect in the text and we are indeed thankful to the Reviewer for his comments. Obviously, the task was complicated by the data limited situation as it is typical for such events. Nevertheless, for the sake of this work the authors collected a vast amount of data which was also a great effort of different teams. Yet,

these results have already been discussed multiple times and accepted by German parties involved in the International Commission for the Protection of the Odra River against Pollution.

Technical terms were changed: "water table" to "water surface", and "velocity coefficient" to "Strickler's coefficient.

It is true that a contradictory and mysterious phrase at line 208 writes: "QNL,in (t), ...preliminarily interpolated from the two neighboring gauge stations (uncertain, to be verified)" but this piece of information, if present (what is the meaning of uncertain, to be verified), does not show up in any other part of the paper.

It was misprinted in the previous version. Thank you very much.

Accordingly, failing to detail this fundamental point, as well other informations partly listed in the following, in my opinion the colored maps of Fig. 11 and 12, have no particular relevance because the overall procedure looks flawed.

We hope that after these fundamental explanations the Reviewer will change his opinion on the numerical approach and modelling results. Further, there are more detailed clarifications following the Reviewer's work:

Follow a list of more particular but important details that show that the paper has not yet been carefully reviewed by the Authors

| Line | text | Observation |
|---|---|---|
| 16 | cauese | cause
corrected |
| 82 | Maximum yield of the weir | The maximum yield of the weir is given in line 81-84, p. 3 |
| 118 | The return period of the flood.. | On the basis of what ? Analysis of Rainfall, maximum discharge ? Measured where ?
The return period is evaluation based on the statistics of the yearly peak flows for the gauge stations. Explanation and reference added to the text. |
| 120 | On the 7th of August at the Ostrózno gauge station, the highest ˙
water level of the flash flood occurred at 16:40. The Ręczyn gauge station was recording the water level until the time of 15:20, and thus until it was destroyed due to | Here, as in many following points, you mention to the existence of gauge stations, but without showing the available data.
A graph should be added with all the available measured level or discharge hydrographs at the relevant stations during the flood. On the some graph the timing of the most important events listed.
A graph presenting recorded water levels will be added. |
| 129 | On the 7th of August, the estimated flood rate was 615 m3s/1 | Delete. Already said at line above
Done |

| 134 | The water level ... | Is there any recording of the water level as a function of time ? It would be important to show the elevation as a function of time and in correspondence the operation of the gates.
There are data for all three closures available until the power supply was working. We reconsider adding a graph to the text. |
|---|---|---|
| 138 | After the water level exceeded the edge of the repaired gate, | What do you mean ? Explain better
This sentence was changed to:
After the water level exceeded the edge of the maintenance gate at the inlet to hydropower channel |
| 142 | which is documented in Sup. 1 | No, in the supplementary materials there are some pictures (where ?) and two maps. No other material is available on the dam Breach
Finally not added due to copyright issues. Corrected. |
| 167 | Radomierzyce through the Mill channel. | Every place that is mentione in the paper must be retracebale on the map. I don't see this place neither in Figure 1 nor 6 which are the ones mentioned so far in the paper. At the same time, regarding the name of the rivers, you must use always the same name (Nysa Łuzycka River and e Lusatian Neisse River are probably the same river) and it must be the one that appears on the map
The Mill channel is not indicated on Figure 1 and 6 due to the scale of the map (for clearance). Instead the Mill channel is indicated on Fig. 10, with a reference in text. |
| 170 | destruction (disintegration) of the buildings | Do you mean collapse ?
Yes. corrected |
| 173 | it flooded the Hagenwerder estate | As at line 167
The Hagenwerder estate is located next to the Pliessnitz mounth. Corrected. |
| 175 | city of Zgorzelec on the Polish side (the peak of the wave in Zgorzelec was at 6:40 UTC) | Here one starts realising that a second flood is superimposed to the dam breach wave but considering that you do not clearly explain this point in advance one is left puzzled at how it is possible that the dam breach wave takes so long to get to this town.
Explanation amended in the text. |
| 203 | To restore | Such a wording was found in a publication.n
We change it e.g. 'to determine' (we noted such a wording in a publication). |

| 207 | Equation 1 | This equation is wrong because it does not include the dV/dt term. In the following text you list dV, that does not appear in the equation but this is another error because dV is a volume and is not dimensionally coherent with discharge Q.
 Thank you very much for this comment. Surely, we have misprinted, the correct expression of Eq.(1) included the term dV/dt is shown in the revised version. |
|---|---|---|
| 213 | dV | dV does not appear in the equation but this is another error because dV is a volume and is not dimensionally coherent with discharge Q.
 as above |
| 215 | Measured discharge | Did somebody actually measure the discharge during the flood ? This is a complex task: how did they do it ?
 The discharge was measured by a team of the hydrometry service of the Institute of Meteorology and Water Management (measurement made from the bridge in Zgorzelec). |
| 218 | This enables the inflows QNL,in(t) and QND(t), while taking into account the additional inputs of the Pliessnitz and Czerwona Woda rivers (which were relatively insignificant), to be found. | In my opinion there are a lot of ways to match the measured discharge with different input hydrographs. You do not discuss this point in sufficient detail.
 Thank you very much for this comment. Using an iterative approach, by trial and error one can determine approximate input hydrographs given the peak timing and high water marks. Additional explanations on this procedure to be added in the corrected text. |
| 231 | Velocity coefficient 1/n | This is what everybody call Strickler's coefficient. By the way in the map in the supplementary file you show Strickler's coeffciect as low as below 2.5. This is actually an unbelievable value: which type of ground cover did you model with this low value ?
 1/n=M is corrected to Strickler coefficient. For building areas the M value was set to 1. |
| 245 | Fig 11 | Why ?
 (see Fig. 11) - removed. |
| 258 | the flooding at 10:00 on August 8, 2010, when the flood peak reached the city of Zgorzelec | From figure 12 one would say between 6 and 9 AM.
 As a result of the conducted simulation, Figure 12 illustrates the flooding at |

| | | 10:00 on August 8, 2010, soon after the flood peak reached the city of Zgorzelec. |
|---|---|---|
| 264 | based on the water level increase in the lake | Having the variation as a function of time would be another important calibration point. But nothing is shown in the paper about this important point
There are no data regarding the variation of the overflow to the Berzdorfer lake; only the total volume is known, as written in l. 211. The overflow volume computed by the program was related to the local water level, this in turn depended on the maximum discharge in the river (including the outflow from the broken Niedów dam) and to the adopted roughness values. Another indication for the choice of the Strickler coefficient was satisfying the timing of the water lever rise in Zgorzelec city. To low or to high values resulted in a faster or delayed flood propagation. |
| 266 | The total volume
of released water due to the dam's failure was equal to 22 million m3 | No, this is false. Due to the dam failure only the volume stored in the reservoir was released,
Right, additional outflow from the reservoir was ca. 5 million m3, the remaining part was the volume of the flood wave coming from upstream |
| 275 | the travel time of the first flood peak from the outflow from the Niedów reservoir to the Zgorzelec gauge station
took about seven hours | This is really strange, considering that the two cross section are probably 10 kms apart. It would imply an average velocity of about 0.4 m/s that is really low for a dam breach flood. This point should be discussed better…
This is not strange given the wide valley topography, relatively low longitudinal slope, meandering river character, vegetation influence, retention increased by two weirs. The model simulations are in agreement with the observation at the Zgorzelec gauge station. This issue will be additionally commented. |

| 285 | A particular feature of the Niedów dam ....

was the fact that the homogenous embankments made of sand and gravel had a concrete facing, which acted as an impermeable barrier. | To be honest I was surprised to hear than an earth dam was totally made with sand with a permeability coefficient of $2.8 \times 10^{-3}$ ms$^{-1}$, that is huge, without any impermeable core. Accordingly, the waterproof coating on the inner side of the embankments was totally mandatory and is certainly not a "particular feature" but a must. Rather, I would have concentrated my discussion on two considerations: 1) the 1/100 year return time for the design discharge of the dam was clearly inadequate. 2) the maintenance of the hydropower station that apparently led to the cut-off of the power supply and so contributed to the disaster, was scheduled without the needed attention to the possible occurrence of a flood in that period of the year.
Thank you very much for this valuable comment. Opis katastrofy został uzupełniony o ten aspect in revised version. |

| Table 1 | Apparently the dynamic of the gates opening is in contradiction with the text where you write that "The crew still tried to open more gates manually from the dam's crest, but were unsuccessful." Accordingly, one would expect that after |

| | |
|---|---|
| | 15:36 the gates stay fixed in their position.

Moreover, if a leve recording is available at Ostrozno it should be plotted as a function of time
Thank you very much for this comment. The gates 1 and 2 were partially manually opened from the crest of the dam. It is clarified in revised version. Ostróżno gauge readings were made up to 16:45, then broken due to access difficulties. Water level hydrographs will be added.
. |
| Table 2 | The peak discharge is a result of your model ? You must specify it
Yes, this value is calculated by the model and will be specified accordingly. |
| Figure 2 | Add ruler for distances
The ruler has been added |
| Figure 6 | You show the state borders (which are pretty unrelevant and should be dropped) but not the border of the catchments
The state borders are removed |
| Figure 10 | Gauge ZgorZelec appears twice. Which is the right one ?
Moreover in the paper all the level/discharge recording at the different gauge stations must be shown as a function of time during the event.
A figure showing water level hydrographs for gauging stations will be added. |